# Home-Based REM Sleep Without Atonia in Patients with Parkinson’s Disease: A Post Hoc Analysis of the ZEAL Study

**DOI:** 10.3390/neurosci7010006

**Published:** 2026-01-03

**Authors:** Hiroshi Kataoka, Masahiro Isogawa, Hitoki Nanaura, Hiroyuki Kurakami, Miyoko Hasebe, Kaoru Kinugawa, Takao Kiriyama, Tesseki Izumi, Masato Kasahara, Kazuma Sugie

**Affiliations:** 1Department of Neurology, Nara Medical University, Kashihara 634-8521, Japan; sevenstar7n@naramed-u.ac.jp (H.N.); kinugawa_kaoru@naramed-u.ac.jp (K.K.); kiri@naramed-u.ac.jp (T.K.); k130739@naramed-u.ac.jp (T.I.); ksugie@naramed-u.ac.jp (K.S.); 2Institute for Clinical and Translational Science, Nara Medical University Hospital, Kashihara 634-8521, Japan; isogawa@naramed-u.ac.jp (M.I.); kurakami@naramed-u.ac.jp (H.K.); hasebe@naramed-u.ac.jp (M.H.); kasa@naramed-u.ac.jp (M.K.)

**Keywords:** parkinson, REM, REM sleep without atonia, sleep, REM sleep behavioral disorder, movement disorder

## Abstract

REM sleep behavioral disorder (RBD) is of increasing interest in Parkinson’s disease (PD). Previous studies exploring the association between REM sleep without atonia (RWA) and clinical PD features or other objective sleep metrics are scarce and have used PSG findings. A mobile electroencephalography (EEG)/electrooculography (EOG) recording system with two channels can objectively measure sleep parameters, including RWA, during natural sleep at home. We investigated whether RWA measured on a portable recording device at home could be associated with clinical PD features or other sleep metrics using baseline data from the ZEAL study. Differences between patients with and without RWA was analyzed using ANCOVA test. REM sleep length was significantly longer in patients with RWA than in those without RWA. A multivariate comparison using ANCOVA showed a significant difference in log-transformed REM sleep duration of patients with RWA after adjustment for potential confounders (adjusted mean difference of 1.203; 95% confidence interval 0.468 to 1.937; *p* = 0.003). The strength of this study was that it evaluated the association between RWA during natural sleep at home and clinical variables as well as other sleep metrics. The major result was that patients with and without RWA did not differ in their clinical variables, and there was no relation between RWA and objective sleep metrics other than REM sleep. The duration of REM sleep may be associated with RWA during natural sleep at home.

## 1. Introduction

Rapid eye movement (REM) sleep behavioral disorder (RBD), distinguished by the absence of muscle atonia during REM sleep and related to dream-enacting behaviors, is of increasing interest in Parkinson’s disease (PD) because idiopathic RBD is acknowledged as a prodromal stage of PD [1]. REM sleep without atonia (RWA) is required for the diagnosis of RBD. The primary polysomnographic (PSG) characteristic of RBD is identified as an intermittent or continuous increase in electromyographic tone or the absence of muscle atonia during REM sleep [2], and RWA is observed in 40–75% of patients with PD [3]. RBD has attracted attention from the clinical perspective as it is related to disorder severity [4] and cognitive decline [5,6]. RWA is closely associated with synucleinopathy [7] and may indicate accelerated motor development in patients [8]. Sleep disturbances, which cause fragmented sleep and difficulty maintaining sleep or falling asleep, are frequent non-motor symptoms of PD [9]. Sleep disorders and RBD are more closely related because RBD is associated with degradation of the brainstem sleep-regulating center [10]. Previous studies exploring the association between RWA and other objective measurements of sleep are scarce and have used PSG findings.

PSG, a premier instrument for the objective evaluation of conventional sleep characteristics, was employed to diagnose RBD in the aforementioned PD patients. Patients undergoing PSG must spend the night in a hospital or sleep institution. During a PSG test, multiple electroencephalography (EEG) and electromyography (EMG) are attached to the scalp and body, which may cause discomfort and make it difficult to evaluate natural sleep. SleepGraph^®^ is a portable, two-channel EEG/electrooculography (EOG) recording device (Proassist Co., Osaka City, Osaka Prefecture, Japan) that provides an inexpensive, objective method to measure sleep parameters at home. When comparing an EEG/EOG recording device with two channels with PSG, validation studies on healthy subjects [11,12] and our validation research on PD patients [13] revealed good correlation between estimated sleep parameters. Recently, to evaluate the safety and efficacy of zonisamide for treating sleep disorders in PD patients, we used the portable EEG/EOG recording system to perform a randomized, single-blind, placebo-controlled trial of ZEAL (Zonisamide’s efficacy for sleep irregularity of PD) [14]. Based on the objective sleep metrics from the portable EEG/EOG recording system, a post hoc analysis of the ZEAL study revealed that psychosis may be associated with nocturnal symptoms of PD that interfere with regular sleep at home [15]. It also emphasized that the two-channel EEG/EOG recording system can measure RWA during natural home-based sleep.

This post hoc analysis of the ZEAL study aimed to investigate whether RWA measured on a portable two-channel EEG/EOG recording device at home could be associated with clinical PD features or other sleep metrics.

## 2. Materials and Methods

### 2.1. Study Design and Participants

This study investigated the association between RWA and clinical PD features or other sleep metrics using baseline data of the ZEAL study, which was enrolled in the Japan Registry of Clinical Trials (jRCTs051200160). Patients with PD who were registered in the ZEAL study conducting at Nara Medical University Hospital, Japan, were analyzed. The ZEAL study evaluated zonisamide’s efficacy for treating sleep disorders in PD patients according to a previously established protocol [16]. Concisely, all PD patients aged more than 41 years who met the diagnostic criteria for PD according to the International Parkinson and Movement Disorder Society (MDS) [17] were included, with the exception of the requirement for decreased striatal uptake of the dopamine transporter as measured by single-photon emission computed tomography (SPECT). Instead, myocardial uptake was reduced on metaiodobenzylguanidine (MIBG) scintigraphy in 37 patients, and striatal dopamine transporter uptake was decreased on SPECT in 25 patients. Each patient indicated the presence of at least one of the following sleep disturbances and achieved a Mini-Mental State Examination score of ≥22: (1) a score greater than 5 on the Japanese version of the 10-item no/yes Sleep Behavior Disorder Screening Questionnaire (RBDSQ), which has a maximum score of 13 [18]; (2) an answer of “sometimes”, “many”, or “so much” on item 2 (“Did you have a bad day at night?”) of the Japanese version of the PD Sleep Scale (PDSS)-2 [19]; or (3) an answer of “sometimes”, “almost none”, or “nothing” on item 1 (“Did you sleep well last week?”) of the Japanese version of the PDSS-2 [19]. All patients were outpatients. Patients with a history of brain surgery or other organic cerebral diseases, comorbidities including severe dyskinesia, major medical history such as malignant syndrome, or those taking both monoamine oxidase (MAO)-B inhibitors and tricyclic antidepressants were eliminated from the study. No patients had periodic limb movements or restless leg syndrome before enrollment, and no patient underwent video PSG for RBD identification. Patients who met the study’s eligibility requirements provided written informed consent. The study protocol was approved by the Nara Medical University Research Ethics Committee.

### 2.2. Mobile Two-Channel EEG/EOG Recording System

The portable recording device consisted of a pair of bipolar EEG and EOG electrodes, and frontal EEG and EOG recordings were captured and used as the receiver [11,12,13]. Fp1 recorded forehead EEG activity relative to the contralateral mastoid process (M2). For EOG recording, two electrodes were placed on the skin over the opposing chin muscles, approximately one centimeter below the eyes. EEG filters of 0.540 and 0.544 were used to record the signals at a sampling frequency of 128 Hz. Amplified and filtered analog data from the electrodes were transformed into digital signals by a 14-bit A/D converter. After that, the patient’s bedside receiver received this digital signal, which was subsequently saved for offline data analysis. Besides sleep stage structure, the AASM criteria were used to calculate other sleep metrics, such as sleep efficiency, total sleep time, wake time following sleep onset, and sleep onset latency [20]. Sleep stage assessment was based on the EEG data from the forehead [12]. EMG activity was considered present when more than 50% of mini-epoch was occupied for 3 s by phasic muscle activity lasting between 0.1 and 5 s, with an amplitude four times greater than the background. The epoch was defined as RWA. An automatic calculation was performed to determine RWA as a proportion of total REM sleep epochs. Our validation study in patients with PD [13] used portable EEG/EOG recording equipment and PSG concurrently in a laboratory setting on the same night. A professional sleep technologist manually and blindly scored both the portable device and the diagnostic PSG. In six out of eight PD patients, the portable device demonstrated moderate-to-high agreement with RWA identification compared with PSG (*p* = 0.686). We used the recorded results of the portable device on two consecutive nights before the zonisamide intervention in a randomized clinical trial of ZEAL.

### 2.3. Other Clinical Evaluations

The Pittsburgh Sleep Questionnaire, PDSS-2, RBDSQ, Hoehn–Yahr stage, MDS Unified Parkinson’s Disease Rating Scale (UPDRS) parts 3 and 4 [21], “psychosis” and “anxiety” items on the MDS non-motor rating scale [22], third edition of the Beck Depression Inventory (BDI) [23], and Parkinson’s Fatigue Scale [24] were among the clinical assessments. A recent assessment and proposal [25] were used as the basis for determining the daily levodopa equivalent dose (LEED). In accordance with earlier studies, the subdomains of the PDSS-2 in Japanese—motor symptoms at night, nocturnal PD symptoms, and disrupted sleep—were evaluated [26].

### 2.4. Statistical Analysis

The means and standard deviations of normally distributed data were displayed, and medians and interquartile ranges were used to display variables with asymmetric distributions (Table 1). Individuals with and without RWA were compared for differences in sleep metrics obtained from the two-channel EEG/EOG recording device and clinical features using the non-parametric Mann–Whitney U test and chi-squared tests (Table 1). Specifically, the difference between patients with and without RWA were analyzed using the analysis of covariance (ANCOVA) test (Table 2), with covariates including age, sex, body mass index (BMI), LEED ≥ 400, disease duration > 109 (median value), Hoehn–Yahr stage, MDS-UPDRS part 3 and part 4, depression defined as BDI score ≥ 14, fatigue defined as Parkinson’s Fatigue Scale ≥ 3.3, “anxiety” and “psychosis” both defined as each score ≥ 2, RBDSQ ≥ 5, motor symptoms during the night on the PDSS, and nocturnal symptoms of PD and disturbed sleep on the PDSS (Table 2). Due to a skewed distribution, REM sleep, nocturnal symptoms of PD on the PDSS, and MDS-UPDRS part 3 were analyzed after natural log transformation (Table 2). SPSS version 24 (IBM Corp., Armonk, NY, USA) was used for all statistical analyses. The ZEAL study data used in the present study were checked for quality by the data manager, monitoring members of the Institute for Clinical and Translational Science, and the principal investigator. Before this, we consulted the manufacturer, Proassist Co., Osaka City, Osaka Prefecture, Japan, to confirm the technical quality of the sleep graph data. Data entry was performed by both the principal investigator and a co-researcher, and consistency was verified.

## 3. Results

Patients with RWA had significantly longer REM sleep durations than patients without RWA (Table 1). A multivariate comparison using ANCOVA showed a significant difference in log-transformed REM sleep of patients with RWA (adjusted mean difference of 0.927 [95% confidence interval: 0.263 to 1.591; *p* = 0.008]) after adjusting for age, sex, and BMI (Table 2). The significant difference was also unchanged (Table 2), independent of age, sex, BMI, LEED, and disease duration (adjusted mean difference of 1.032 [95% confidence interval: 0.319 to 1.745; *p* = 0.006]), or independent of age, sex, BMI, LEED, disease duration, Hoen–Yahr stage, log-transformed MDS-UPDRS part 3, and MDS-UPDRS part 4 (adjusted mean difference of 1.067 [95% confidence interval: 0.335 to 1.799; *p* = 0.006]), or independent of age, sex, BMI, LEED, disease duration, Hoen–Yahr stage, log-transformed MDS-UPDRS part 3, MDS-UPDRS part 4, depression, fatigue, anxiety, psychosis, RBDSQ, motor symptoms during the night, log-transformed nocturnal PD symptoms, and sleep disturbances on the PDSS (adjusted mean difference of 1.203 (95% confidence interval: 0.468 to 1.937; *p* = 0.003) (Table 2). Compared with patients who did not have RWA or RBDSQ ≥ 5, Hoehn–Yahr stage (*p* = 0.039) and MDS-UPDRS part 3 (*p* = 0.037) showed a significant increase in patients with RWA without RBDSQ ≥ 5 (Table 3), and the score for motor symptoms at night on the PDSS showed a significant increase in patients with both RWA and RBDSQ ≥ 5 (*p* = 0.038) (Table 3). Between patients with RWA without RBDSQ ≥ 5 and those with both RWA and RBDSQ ≥ 5, clinical PD characteristics and objective sleep parameters did not differ except for MDS-UPDRS part 3 (Table 3). RWA in 15 patients with both RWA and RBDSQ ≥ 5 was 17.7 ± 3.1% (median 17.4, range 1.2 to 39.1%), while 10 patients with RWA who did not have RBDSQ ≥ 5 had RWA of 27.0 ± 7.6% (median 20.6, range 4.4 to 84.3%) (Table 4), and there was no difference between the two groups (*p* = 0.428). Patients with RWA did not show a typical clinical history of RBD or dream-enacting behavior in the prodromal stage of PD. Three patients with RWA received etizolam.

## 4. Discussion

The strength of this research was that it evaluated the association between RWA during natural sleep at home and clinical variables as well as other sleep metrics. The major result was that patients with and without RWA did not differ in their clinical variables, and no association was found between RWA and objective sleep metrics other than REM sleep.

RBD may contribute to more severe PD symptoms compared to those without RBD symptoms [27]. Patients with higher RWA also had worse PD severity [28]. RWA has been associated with Hoehn–Yahr stage, PD duration, LEED, UPDRS III, and Montreal Cognitive Assessment (MoCA) score [29]. The development of rigidity symptoms in the neck and limbs is more significantly impacted by REM sleep disruption, which is indicated by high tonic RWA density [30]. In the current study, no association was found between RWA and PD severity. Although the exact reason for this finding remains unknown, it is likely related to the cross-sectional design of this study. A correlation may exist between RWA and clinical severity of PD if the group is monitored longitudinally. Another reason may be that this study used a portable two-channel EEG/EOG recording device, in contrast to the aforementioned studies that used PSG.

Additionally, clinical PD characteristics were evaluated in patients with RWA who had probable RBD, defined as RBDSQ ≥ 5, and in those who did not. Similarly, a previous study evaluated two groups of PD patients: those with dream-enacting behavior and RWA positive for RBD diagnosis, and PD patients who did not experience dream-enacting behavior but showed a significant amount of RWA [4]. Both groups had a higher Hoehn–Yahr stage, UPDRS part 3, LEED, lower MoCA scores, a higher proportion of N1 sleep, and a lower proportion of N2 sleep than patients with normal REM sleep. Both groups exhibited similar clinical features and PSG parameters. In the current study, compared with patients without RWA or probable RBD, patients with RWA without probable RBD had greater increases in Hoehn–Yahr stage and MDS-UPDRS part 3. PDSS-2 showed greater impairment in nocturnal motor symptoms in patients with both RWA and probable RBD. Between patients with RWA who had positive and negative RBDSQ ≥ 5, differences in PD features and sleep metrics were not significant, consistent with previous findings [4]. PD patients with a typical clinical history of RBD and dream-enacting behaviors exhibited typical behaviors of RBD during video PSG. However, some PD patients exhibit elevated RWA without dream-enacting behavior on the basis of their clinical history or PSG videos [31], and this would be regarded as concomitant subclinical RBD. Confirmation of clinical PD characteristics in both definite and subclinical RBD will be necessary for a larger study.

Although prior research has examined PSG variables between PD patients with and without RBD, precise alternations in objective sleep measures have not been determined. In one study, PSG sleep efficiency was greater in PD patients with RBD, defined by RWA and intricate movements during REM sleep, than in those without RBD [32]. In the present study, REM sleep differed between PD patients with and without RWA, regardless of potential confounders, when assessed on a mobile two-channel EEG/EOG recording device, similar to previous PSG studies [3,33]. Two PSG studies showed that REM sleep, measured as % of total sleep time, was greater in PD patients with RBD than in those without RBD [32,33]. According to a meta-analysis of studies comparing the differences in PSG sleep metrics between PD patients with and without RBD, PD patients with RBD had a significantly higher REM sleep percentage [29]. Phasic RWA has positive correlation with REM duration [29]. Disease-unique progressive degenerative processes in brainstem areas regulating REM sleep in patients with idiopathic RBD cause RWA to increase over time [34].

Because it allows for the objective measurement of RWA and direct visualization of dream-enacting activities, PSG is still the gold standard for diagnosing RBD. The urgent need for home-based alternatives is highlighted by its high cost, restricted accessibility, and impracticability for large-scale screening. To address this, we employed a recording device with two channels (SleepGraph^®^, Proassist Co., Japan) comprising a pair of bipolar EEG and EOG electrode leads with a receiver that records frontal EEG and EOG signals [12,13]. Beyond conventional sleep metrics such as sleep efficiency, total sleep time, wake time after sleep onset, and sleep onset latency, this recording system allows accurate detection of REM sleep, disorders during REM sleep, and RWA. Technologies for home-based RBD detection using such recording systems are rapidly evolving. Novel approaches, including temporary tattoo electrode arrays or soft electrode arrays with Bluetooth-enabled data acquisition units, are able to record EEG, EOG, and EMG signals for sleep monitoring at home [35,36]. Compared with laboratory-based PSG-confirmed RBD with RWA detection, a validation study demonstrated that simultaneous single-night recordings in each PD patients and healthy controls achieved a sensitivity of 85.7% and specificity of 58.3% for RWA detection in the sleep laboratory, and 100% sensitivity with 89.47% specificity during home-based recordings. Similarly, the Sleep Profiler wearable device [37], a patient-applied in-home sleep monitor, has shown accuracy comparable to SleepGraph^®^. An in-laboratory validation of automated RWA detection in RBD patients reported sensitivity and specificity of 88% and 81% (chin), 93% and 81% (arm), and 90% and 86% (chin or arm), respectively, closely approximating expert visual scoring. While its portability and cost-effectiveness make these devices appropriate tools for home-based diagnostics, sensitivity may be compromised by restricted electrode arrangements [38]. The reliability of video-based and biopotential recordings may also be influenced by environmental conditions such as lighting, location, and ambient noise [38]. For SleepGraph^®^, electrode displacement caused by body movements (e.g., turning in bed) represents a technical limitation; this can be mitigated by securing the head with a net and fastening the leads. Furthermore, operating the device—wearing electrodes, handling the receiver, and inserting the electronic medium that records the measurement data—can be challenging for patients with PD. Consequently, proficiency typically requires several nights of practice rather than a single-night measurement. To improve patient understanding, we used photographs showing the equipment components and created a new instruction manual to explain the procedures and precautions for using the equipment. We also explained this to the patients and their families or caregivers and sought their cooperation. However, a systematic review and network meta-analysis also mentioned that PSG-derived sleep alterations may not only improve differential diagnosis but also offer prognostic and therapeutic insights across cognitive decline, including mild cognitive impairment and PD dementia, compared with healthy controls [39]. Cognitive decline itself can be a risk for proper measurement because the portable two-channel device must be worn correctly by the patient. A sensitive and scalable approach for identifying motor abnormalities associated with RBD has been proposed through video-based RBD detection. This approach employs a machine learning classifier using a simple 2D camera [40] and, in some cases, has outperformed more complex video-based 3D systems [41]. This contactless system leverages machine learning to evaluate and track small body movements during REM sleep. However, existing studies have been limited to clinical sleep laboratories and lack validation in-home environments. Moreover, these video-based techniques do not quantify RWA [38]. Portable EEG/EOG recording systems enable evaluation of natural sleep in the home setting. Their portability allows objective assessment of both conventional sleep parameters and RBD. Portable recording systems, including EEG/EOG/EMG-based devices [12,13,36] and the Sleep Profiler [37], have demonstrated strong potential for automatic RWA detection, showing moderate-to-high correlations with RWA measurements obtained from PSG. More recently, a case of RBD undetected in two overnight PSG studies was successfully identified using a consumer-level sleep-monitoring wearable device, highlighting the emerging role of wearable technology in RBD detection and its potential to improve early diagnosis and patient care [42]. Furthermore, a fully automated actigraphy-based model using high-resolution wrist actigraphy (Axivity AX6) and machine learning demonstrated the generalizability of a fully automated algorithm for detecting iRBD, independent of different ethnic cohorts and wrist-worn accelerometer devices [43]. Idiopathic RBD is well recognized as a prodromal stage of neurodegenerative diseases such as PD. If early disease-modifying therapy at the prodromal stage is feasible, then therapeutic biomarkers will be required. However, PSG currently requires overnight hospitalization, which is unlikely for many patients with suspected RBD. If home-based RBD technology is established, it will be easier to examine RBD at home in the outpatient setting, and it can identify a large number of patients with idiopathic RBD. Further technological developments are needed to optimize home-based RBD screening and overcome the limitations of current approaches.

This study’s strength was the utilization of readily available at-home equipment to objectively assess multiple components of sleep, including RWA. The results of the current study may have been affected by the limited sample size, which limited the statistical power and generalizability of the results. PD patients with RBD had greater periodic limb movements during sleep index scores than those without RBD, which are factors affecting sleep disorders [29,32]; however, this index could not be measured using a portable device with two channels. Although PSG was not performed in the current study, periodic limb movements were not observed during routine clinical examinations. We employed the RBDSQ instead of time-synchronized videos to identify dream-enacting behaviors. The RBDSQ showed high sensitivity, specificity, and reliability as a screening tool for idiopathic RBD in an elderly Japanese population [18]. However, reliance on the RBDSQ instead of video-confirmed dream-enacting behaviors may lead to misclassification. The current study did not find a relationship between RWA and clinical PD variables, likely due to the cross-sectional nature of the study. Longitudinal assessment would be necessary to clarify the relationships between RWA and PD progression.

## 5. Conclusions

Utilizing a two-channel EEG/EOG recording device, we were unable to identify any clinical PD features in patients with RWA. However, the duration of REM sleep may be associated with RWA during natural sleep at home.

## Figures and Tables

**Table 1 neurosci-07-00006-t001:** Difference in clinical characteristics between patients with and without REM sleep without atonia (RWA).

	With RWA *n* = 27	Without RWA *n* = 14	*p*	Effect Size
age, mean, years	72.4, 6.6	73.1, 7.6	0.923	0.02
men, *n*	18 (66)	7 (50)	0.332	0.162
BMI, mean	21.9, 3.3	21.1, 3.4	0.564	0.09
disease duration (months), median, IQR	112 (84, 135)	96 (73, 160)	0.544	0.10
levodopa daily equivalent dose (mg/day), median, IQR	709 (474, 1237)	537 (328, 1148)	0.211	0.20
LEED ≥ 400	22 (81)	9 (64)	0.267	0.190
Mini-Mental State Examination, mean	28.4, 1.2	28.5, 1.9	0.989	0.0
Hoehn–Yahr stage, mean	2.5, 0.9	2.2, 0.7	0.506	0.10
MDS-UPDRS part 3, mean	24.8, 9.2	23.1, 15.3	0.265	0.17
log-transformed MDS-UPDRS part 3, mean	3.18, 0.40	2.95, 0.78	0.265	0.17
MDS-UPDRS part 4, mean	5.3, 3.1	4.0, 3.9	0.199	0.20
PDSS-2, mean	22.8, 8.9	22.5, 8.7	0.783	0.04
motor symptoms at night, mean	6.5, 3.7	5.8, 3.4	0.477	0.12
nocturnal PD symptoms, median, IQR	5.5 (2.0, 8.0)	5.3 (2.0, 8.5)	0.967	0.01
log-transformed nocturnal PD symptoms, mean	1.72, 0.64	1.73, 0.64	0.967	0.01
disturbed sleep, mean	10.5, 3.1	11.0, 2.7	0.609	0.08
RBDSQ, mean	5.7, 2.9	6.3, 3.5	0.782	0.04
RBDSQ ≥ 5	15 (55)	8 (57)	>0.999	0.015
Beck Depression Inventory, mean	12.9, 6.4	15.3, 7.6	0.205	0.20
presence of depression, *n*	24 (88)	11 (78)	0.393	0.222
Parkinson’s Fatigue Scale, mean	49.7, 15.6	50.5, 13.2	0.815	0.04
presence of fatigue, *n*	13 (48)	5 (35)	0.52	0.119
MDS non-motor rating scale				
anxiety, mean	6.9, 5.7	6.5, 5.1	0.912	0.02
presence of anxiety, *n*	22 (81)	11 (78)	>0.999	0.035
psychosis, median	1.3 (0, 3.0)	1.0 (0, 2.0)	0.563	0.09
presence of psychosis, *n*	12 (44)	6 (42)	>0.999	0.015
objective sleep parameters				
sleep efficiency (%), median, IQR	89.4 (78.6, 94.9)	86.6 (82.5, 91.2)	0.858	0.03
total sleep time (TST) (min), mean	338.3, 92.0	319, 111.7	0.869	0.03
wake time after sleep onset (WASO) (min), median, IQR	44.5 (19.5, 83.5)	50.5 (35.3, 69.3)	0.67	0.07
sleep onset latency (SOL) (min), median, IQR	21.0 (10.5, 34.5)	16.8 (11.8, 24.1)	0.621	0.08
REM sleep (min), median, IQR	50.2 (31.5, 80.5)	30.0 (6.6, 50.8)	0.014 ^†^	0.38
log-transformed REM sleep, mean	3.90, 0.73	2.97, 1.29	0.014 ^†^	0.38
non-REM sleep (min), median, IQR	265.0 (225.0, 320.0)	297.3 (255.5, 354.1)	0.161	0.22
deep sleep (N3) time (min), mean	70.6, 26.9	70.4, 26.9	0.731	0.05

BMI: body mass index, PDSS: Parkinson’s Disease Sleep Scale-2 Japanese version, LEED: levodopa daily equivalent dose, RBDSQ: Sleep Behavior Disorder Screening Questionnaire Japanese version, MDS-UPDRS: Movement Disorder Society Revision of the Unified PD Rating Scale, REM: Rapid Eye Movement, data are reported as mean (standard deviation), median (IQR), or number (%), ^†^: *p* < 0.05.

**Table 2 neurosci-07-00006-t002:** Association between REM sleep without atonia (RWA) and REM sleep.

Log-Transformed REM Sleep	Without RWA	With RWA							
		Difference	95% CI		Ratio (Exp (Difference))	95% CI		*p*	Effect Size
model 1	ref.	0.927	0.263	1.591	2.527	1.301	4.909	0.008	0.207
model 2	ref.	1.032	0.319	1.745	2.807	1.376	5.726	0.006	0.238
model 3	ref.	1.067	0.335	1.799	2.907	1.426	6.044	0.006	0.323
full-adjusted ^†^	ref.	1.203	0.468	1.937	3.331	1.597	6.938	0.003	0.570

Model 1: adjusted for age, gender, and BMI. Model 2: adjusted for age, gender, BMI, LEED ≥ 400, and disease duration > 109. Model 3: adjusted for age, gender, BMI, LEED ≥ 400, disease duration > 109, Hoen–Yahr stage, log-transformed MDS-UPDRS part 3, and MDS-UPDRS part 4. ^†^: adjusted for age, gender, BMI, LEED ≥ 400, disease duration > 109, Hoen–Yahr stage, log-transformed MDS-UPDRS part 3, MDS-UPDRS part 4, depression, fatigue, anxiety, psychosis, RBDSQ ≥ 5, motor symptoms at night on PDSS, log-transformed nocturnal PD symptoms on PDSS, and disturbed sleep on PDSS. REM: rapid eye movement; BMI: body mass index; PDSS: Parkinson’s Disease Sleep Scale-2 Japanese version; LEED: levodopa daily equivalent dose; RBDSQ: Sleep Behavior Disorder Screening Questionnaire Japanese version; log-transformed MDS-UPDRS: Movement Disorder Society Revision of the Unified PD Rating Scale.

**Table 3 neurosci-07-00006-t003:** Difference in clinical variables of patients with REM sleep without atonia (RWA) between patients with and without RBDSQ ≥ 5.

	Patients Without RWA or RBDSQ ≥ 5 (Group A), *n* = 16	Patients with RWA Without RBDSQ ≥ 5 (Group B), *n* = 10	*p* (Group A vs. B)	Patients with Both RWA and RBDSQ ≥ 5, *n* = 15 (Group C)	*p* (Group A vs. C)	*p* (Group B vs. C)
age, mean, years	73.6, 7.3	72.9, 7.9	0.853	71.6, 5.9	0.475	0.605
men, *n*	9	6	>0.999	10	0.716	>0.999
BMI, mean	21.0, 3.4	21.9, 4.5	0.752	22.1, 2.6	0.323	0.892
disease duration (months), median, IQR	96 (71, 160)	101 (72, 124)	0.731	117 (108, 135)	0.373	0.311
levodopa equivalent daily dose (mg/day), median, IQR	519 (341, 988)	950 (456, 1249)	0.14	760 (474, 1058)	0.166	0.892
LEED ≥ 400	10	8	0.42	13	0.22	>0.999
Mini-Mental State Examination, mean	28.6, 1.8	28.3, 1.4	0.399	29.0, 1.2	0.506	0.261
Hoehn–Yahr stage, mean	2.3, 0.7	3.0, 1.0	0.039 ^†^	2.3, 0.9	0.908	0.091
MDS-UPDRS part 3, mean	22.4, 14.9	29.0, 6.7	0.037 ^†^	23.1, 9.7	0.579	0.041 ^†^
MDS-UPDRS part 4, mean	4.5, 3.8	4.8, 2.4	0.691	5.5, 3.8	0.473	0.723
PDSS-2, mean	22.5, 8.2	19.2, 8.5	0.509	25.3, 9.2	0.417	0.103
motor symptoms at night, mean	5.4, 3.5	5.5, 3.9	0.937	7.8, 3.3	0.038 ^†^	0.144
nocturnal PD symptoms, median, IQR	5.3 (2.3, 8.0)	5.0 (2.8, 7.0)	0.691	5.7 (2.0, 10.0)	0.905	0.765
disturbed sleep, mean	11.4, 2.8	8.8, 2.7	0.041 ^†^	11.2, 3.1	0.751	0.08
RBDSQ, mean	6.1, 3.4	3.1, 0.7	0.015 ^†^	7.8, 2.5	0.061	<0.001
RBDSQ ≥ 5	8 *	0	0.009 ^†^	15	0.002 ^†^	<0.001
Beck Depression Inventory, mean	15.6, 7.3	12.9, 7.0	0.355	13.0, 6.6	0.342	0.935
presence of depression, *n*	13	9	>0.999	13	>0.999	>0.999
Parkinson’s Fatigue Scale, mean	6.1, 5.0	8.1, 5.2	0.356	6.7, 6.3	0.428	0.495
presence of fatigue, *n*	6	3	>0.999	9	0.289	0.226
MDS non-motor rating scale						
anxiety, mean	6.1, 5.0	8.1, 5.2	0.396	6.7, 6.3	0.904	0.428
presence of anxiety, *n*	12	10	0.136	11	>0.999	0.125
psychosis, median	1.0 (0, 2)	0.8 (0, 2.5)	0.694	1.6 (0, 3.0)	0.403	0.367
presence of psychosis, *n*	7	3	0.683	8	0.724	0.414
objective sleep parameters						
sleep efficiency (%), median, IQR	84.9 (75.9, 90.0)	88.3 (75.7, 96.5)	0.429	89.8 (81.8, 94.9)	0.26	0.892
total sleep time (TST) (min), mean	311.4, 106.1	332.5, 98.0	0.635	352.9, 91.7	0.363	0.723
wake time after sleep onset (WASO) (min), median, IQR	57.0 (39.9, 103.8)	41.5 (16.3, 108.6)	0.527	42.0 (19.0, 63.0)	0.149	0.683
sleep onset latency (SOL) (min), median, IQR	18.8 (12.5, 18.8)	35.0 (7.1, 97.0)	0.384	17.5 (10.5, 28.0)	0.489	0.129
REM sleep (min), median, IQR	32.6 (9.9, 53.3)	62.3 (31.0, 83.1)	0.058	43.5 (30.5, 84.5)	0.079	0.935
non-REM sleep (min), median, IQR	296.8 (218.0, 346.0)	261.8 (209.4, 323.0)	0.343	270.5 (239.5, 330.5)	0.843	0.397
deep sleep (N3) time (min), mean	68.3, 33.4	65.1, 17.5	0.752	76.7, 32.2	0.678	0.461

BMI: body mass index; PDSS: Parkinson’s Disease Sleep Scale-2 Japanese version; LEED: levodopa daily equivalent dose; RBDSQ: Sleep Behavior Disorder Screening Questionnaire Japanese version; MDS-UPDRS: Movement Disorder Society Revision of the Unified PD Rating Scale; REM; rapid eye movement, *: eight patients had RBDSQ ≥ 5, but there was no RWA, data are reported as mean (standard deviation), median (IQR), or number (%), ^†^: *p* < 0.05.

**Table 4 neurosci-07-00006-t004:** Proportion of REM sleep without atonia (RWA) and the 10-item no/yes Sleep Behavior Disorder Screening Questionnaire Japanese version (RBDSQ).

	RBDSQ	RWA (%)
Patients with both RWA and RBDSQ ≥ 5		
Pt. 1	5	39.1
Pt. 2	6	29.7
Pt. 3	11	29.1
Pt. 4	10	14.0
Pt. 5	8	12.8
Pt. 6	6	1.2
Pt. 7	11	14.1
Pt. 8	8	3.3
Pt. 9	5	21.1
Pt. 10	12	21.7
Pt. 11	10	24.1
Pt. 12	6	2.3
Pt. 13	6	4.1
Pt. 14	5	17.4
Pt. 15	8	32.1
Patients with RWA but without RBDSQ ≥ 5		
Pt. 16	3	7.2
Pt. 17	3	18.3
Pt. 18	4	84.3
Pt. 19	2	12.2
Pt. 20	3	22.9
Pt. 21	3	51.4
Pt. 22	2	4.4
Pt. 23	4	26.0
Pt. 24	3	17.5
Pt. 25	4	25.7

RWA: REM sleep without atonia, RBD: rapid eye movement (REM) sleep behavioral disorder.

## Data Availability

All data used to write this manuscript can be reviewed upon reasonable request, and further enquiries can be directed to the corresponding author. Although the data supporting the study’s conclusions are not publicly available, they can be obtained upon request from the corresponding author.

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
