# Peer review of "Home-Based REM Sleep Without Atonia in Patients with Parkinson’s Disease: A Post Hoc Analysis of the ZEAL Study"

_neurosci, 2026, doi:10.3390/neurosci7010006_

Round 1

Reviewer 1 Report

Comments and Suggestions for Authors

Peer-Review Report

Title: Home-Based REM Sleep Without Atonia in Patients with Parkinson’s Disease: A Post-Hoc Analysis of the ZEAL Study

Summary of the Manuscript

The study investigates the association between REM sleep without atonia (RWA) and clinical variations in Parkinson’s disease (PD) using a home-based two-channel EEG/EOG recording system. It is a post-hoc analysis of the ZEAL trial, which originally evaluated zonisamide’s efficacy for sleep disorders in PD. The authors report that RWA correlates with longer REM sleep duration, but not with other sleep metrics or clinical PD severity.

Major Strengths

Novelty: The use of a portable EEG/EOG device for home-based RWA detection could be innovative and addresse the limitations of polysomnography (PSG).

Clinical Relevance: Highlights the potential for scalable, cost-effective RBD screening in PD patients.

Methodological Rigor: ANCOVA models with multiple covariates strengthen the validity of the findings.

Major Concerns

Sample Size: The study includes only 41 patients (27 with RWA, 14 without), which limits statistical power and generalizability.

Cross-Sectional Design: Prevents assessment of longitudinal relationships between RWA and PD progression.

RBD Diagnosis: Reliance on RBDSQ instead of video-confirmed dream-enacting behaviors may lead to misclassification.

Device Limitations: The portable system cannot measure periodic limb movements, which are relevant to sleep disorders in PD.

Specific Comments

Introduction:

- Well-written, but could better emphasize why home-based RWA detection is clinically urgent

 -The sentence “RBD worsens cognitive function and advances more rapidly than PD without RBD” must be better clarified and corroborated by the literature.

-Clarify “More severe RBD may be indicated by a higher level of RWA” (how is this measured and in which case).

Methods:

- It is not clear if patients underwent videopsg for RBD identification or not before being registered with the two channels

-The reported automatic detection of RWA needs a clinical validation in this type of subject.

--The authors should better explain how RWA is calculated.

Results: Tables are comprehensive, but consider adding effect sizes for clinical variables.

Discussion: Appropriately addresses limitations, but could expand on implications for future wearable technologies.

References: Up-to-date

Statistical Analysis

ANCOVA models are appropriate, but the rationale for log-transforming REM sleep should be explained more clearly.

Report confidence intervals for all key comparisons in the main text, not only in tables.

Ethical Compliance

-IRB approval and informed consent are documented. No conflicts of interest reported.

-but no number of CE is reported

Recommendation

Major Revision

Justification: The manuscript is promising but requires clarification on device validation, an expanded discussion on clinical implications, and acknowledgment of potential biases due to the reliance on RBDSQ.

Suggested Actions for Authors

The RBD diagnosis criteria need video polysomnography, either at home or in a lab.

Provide validation data for the portable EEG/EOG device compared to videoPSG.

Discuss how electrode displacement and patient handling issues were mitigated.

Include effect sizes and confidence intervals for all clinical comparisons.

Expand discussion on future directions and limitations for home-based RBD detection.

Author Response

Responded to the reviewer’s suggestions

Reviewer 1

Major Concerns

Sample Size: The study includes only 41 patients (27 with RWA, 14 without), which limits statistical power and generalizability.

 We agreed with the reviewer’s suggestion. We expanded the limitation regarding the small sample in the Discussion.

Cross-Sectional Design: Prevents assessment of longitudinal relationships between RWA and PD progression.

 We largely agreed with the reviewer’s suggestion. The current study did not found between RWA and clinical PD variable because of cross-sectional study, and the longitudinal assessment would identify the relationships between RWA and PD progression. We added it in the Limitation of the text.

RBD Diagnosis: Reliance on RBDSQ instead of video-confirmed dream-enacting behaviors may lead to misclassification.

 We largely agreed with the reviewer’s suggestion again. We added the Limitation about reliance of RBDSQ in the Discussion.  

Device Limitations: The portable system cannot measure periodic limb movements, which are relevant to sleep disorders in PD.

We agreed with the reviewer’s suggestion. All patients did not undergo PSG, but had not periodic limb movements or restless leg syndrome on the clinical routine examination before being registered in the present study. The information was added in the methods and we enhanced the limitation.

Specific Comments

Introduction:

- Well-written, but could better emphasize why home-based RWA detection is clinically urgent

 According to the reviewer’s suggestion, we emphasized the clinical reason for home-based RWA detection in the Introduction.

 -The sentence “RBD worsens cognitive function and advances more rapidly than PD without RBD” must be better clarified and corroborated by the literature.

-Clarify “More severe RBD may be indicated by a higher level of RWA” (how is this measured and in which case).

Thanks for your suggestion. We repeatedly carefully read the reference, and noticed that PD who had RBD and mild cognitive impairment at baseline showed a rapid progression in cognition after 4 to 4.5 years and that RWA is increasing according to severity of Yahr stage or UPDRS. While, other two reviewers pointed that the introduction was too long and repeated known information about RBD and Parkinson’s disease. So we modified largely this part of the Introduction.

Methods:

- It is not clear if patients underwent videopsg for RBD identification or not before being registered with the two channels

 According to the reviewer’s suggestion, we added this information.

-The reported automatic detection of RWA needs a clinical validation in this type of subject.

 As suggested, we already performed the validation study between PSG and the two channel EEG/EOG recording device in patients with Parkinson’s disease as described previously (reference number 13). The portable device demonstrated moderate-to-high agreement with RWA identification as compared to PSG (p value: 0.686). However, we did not perform PSG in the enrolled patients in the present post-hoc analysis. Instead, we added the detailed results of our previous validation study in the Methods.

--The authors should better explain how RWA is calculated.

  As described in the method of the mobile two-channel EEG/EOG recording system, when chin electromyography activity, defined as the duration of phasic muscle activity lasting 0.1–5 seconds with an amplitude four times greater than that of the background, is occupied by more than 50% for mini epochs for 3 s, the epoch was defined as RWA. The ratio of RWA to total REM sleep was calculated automatically.

Results:

Tables are comprehensive, but consider adding effect sizes for clinical variables.

 According to the reviewer’s suggestion, we added the effect sizes for clinical variables.

Discussion:

Appropriately addresses limitations, but could expand on implications for future wearable technologies.

According to the reviewer’s suggestion, we expanded this discussion about the implications of future wearable technologies in 5th paragraph of the Discussion.

References: Up-to-date

 According to the reviewer’s suggestion, we up-to-dated references by adding two references (number 39 and 43).

Statistical Analysis

ANCOVA models are appropriate, but the rationale for log-transforming REM sleep should be explained more clearly.

As suggested, we log-transformed the REM sleep since REM sleep shows a skewed distribution. This is already described in the “statistical analysis” of the Methods.

Report confidence intervals for all key comparisons in the main text, not only in tables.

According to the reviewer’s suggestion, we added all confidence intervals in the Result.

Ethical Compliance

-IRB approval and informed consent are documented. No conflicts of interest reported.

-but no number of CE is reported

  According to the reviewer’s suggestion. We already described “Conflict of Interest Statement” in “Statement of Ethics”of the text (The authors report no conflicts of interest related with the present manuscript). We added the number (CRB number : nara0020) approved by Ethical Compliance about the randomized clinical trials ZEAL (the baseline data of ZEAL was used in the present study). Certified Review Board (CRB) is a committee established under the Clinical Research Act to properly review specific clinical research from the perspectives of ethical and scientific validity. Dose CE means EC (Ethical Compliance)?

Recommendation

Major Revision

Justification: The manuscript is promising but requires clarification on device validation, an expanded discussion on clinical implications, and acknowledgment of potential biases due to the reliance on RBDSQ.

Thanks for your suggestions. As for clarification on device validation, we added the size effect and repeatedly performed the overall statistical analysis of the present study. Also, we expanded the discussion about clarification on home-based devise in 5th paragraph of the Discussion. Regarding the clinical implications, we expanded this discussion in 5thparagraph of the Discussion. About the potential bias of RBDSQ, we added it as a Limitation of 6th paragraph of the Discussion.

Suggested Actions for Authors

The RBD diagnosis criteria need video polysomnography, either at home or in a lab.

We agreed with the reviewer’s suggestion. We did not perform PSG in all enrolled patients in the present post-hoc analysis. However, we already performed the validation study between PSG and the two channel EEG/EOG recording device in patients with Parkinson’s disease as our described previously (reference number 13). The portable device demonstrated moderate-to-high agreement with RWA identification as compared to PSG (p value: 0.686). we added the detailed results of our previous validation study in the Methods.

Provide validation data for the portable EEG/EOG device compared to videoPSG.

As suggested, we performed the validation study between PSG and the two channel EEG/EOG recording device in patients with Parkinson’s disease as described previously (reference number 13). The portable device demonstrated moderate-to-high agreement with RWA identification as compared to PSG (p value: 0.686). However, we performed PSG in the enrolled patients in the present post-hoc analysis. Instead, we added the detailed results of our previous validation study in the Methods.

Discuss how electrode displacement and patient handling issues were mitigated.

As for the electrode displacement, this already described in the 5th paragraph of the Discussion. For SleepGraph®, electrode displacement caused by body movements (e.g., turning in bed) represents a technical limitation; this can be mitigated by securing the head with a net and fastening the leads. Regarding patient handling issues, to make it easier for patients to understand, we used photographs that included the details of the equipment, and a newly created instruction manual to explain the procedures and precautions for using the equipment. We also explained this to the patients and their families or caregivers, and asked for their cooperation. We added the latter in the 5th paragraph of the Discussion.

Include effect sizes and confidence intervals for all clinical comparisons.

 According to the reviewer’s suggestion, we added the effect sizes for clinical variables. About confidence intervals, we used Mann–Whitney U test. Also. we noticed that the presence of depression is defined as BDS score≥ 14, and that the MDS-UPDRS part 3 does not normally distributed. Again, we re-performed the statistical analysis including ANCOVA testing. We modified the text and tables.

Expand discussion on future directions and limitations for home-based RBD detection.

 According to the reviewer’s suggestion, we expanded them for home-based RBD detection in the 5th paragraph of the Discussion.

Reviewer 2 Report

Comments and Suggestions for Authors

This study does not contain any novelty, just provides the statistical analysis with the ANCOVA test. Also, it is concluded that no association was found in the current study between RWA and the severity of PD.

This study does not validate the home-based RWA measurements against full polysomnography (PSG), which is the clinical standard. Without PSG comparison, the accuracy of RWA detection and the reliability of the reported REM duration differences cannot be confirmed.

The paper does not include deeper statistical evaluations such as ANOVA, effect size estimates, or post-hoc comparisons. This limits the ability to determine whether observed group differences are statistically meaningful beyond simple correlations. Authors should provide these evaluations.

The study just included 41 participants. The sample set is small for the number of covariates used in ANCOVA. This may lead to overfitting and reduce the statistical robustness of the conclusions. How does overfitting reduce here?

The study design does not allow causal inference. Therefore, statements linking RWA to disease severity should be softened or rephrased as associations, not predictors.

The introduction repeats known information about RBD and Parkinson’s disease.
Authors could shorten it and focus on the problem addressed by the study would improve clarity.

In Methods, need more clarification about the following:

  • Provide exact criteria and thresholds for RWA scoring.
  • Clarify how many nights of recording were used.
  • Describe data quality checks and signal rejection procedures.

The figure or architecture would benefit from clearer labelling and consistent formatting. It should be provided in the methods and results sections.

Abbreviations are not given properly.

Comments on the Quality of English Language

Some sentences are long and complex. Minor English editing is required for a smoother flow and improved readability.

Author Response

Responded to the reviewer’s suggestions

Reviewer 2

This study does not contain any novelty, just provides the statistical analysis with the ANCOVA test. Also, it is concluded that no association was found in the current study between RWA and the severity of PD.

 Thanks for your important suggestion. The identification for RWA needs PSG which requires attaching many wires to the subject. Objective methods for detecting RWA during natural sleep at home are very limited and still in development as more recent review (refence number 38). As part of this effort, we have attempted to detect RWA using the portable device with two channels, and this is the first time that RWA obtained from the two-channel device has been analyzed to examine its relationship between RWA and clinical features or objective sleep parameters. As a result, it was found to be related to REM sleep length. We believe these are novelty of the present study.

This study does not validate the home-based RWA measurements against full polysomnography (PSG), which is the clinical standard. Without PSG comparison, the accuracy of RWA detection and the reliability of the reported REM duration differences cannot be confirmed.

We agreed with the reviewer’s suggestion. We did not perform PSG in all enrolled patients in the present post-hoc analysis. However, we already performed the validation study between PSG and the two channel EEG/EOG recording device in patients with Parkinson’s disease as our described previously (reference number 13). The portable device demonstrated moderate-to-high agreement with RWA identification as compared to PSG (p value: 0.686). we added the detailed results of our previous validation study in the Methods.

The paper does not include deeper statistical evaluations such as ANOVA, effect size estimates, or post-hoc comparisons. This limits the ability to determine whether observed group differences are statistically meaningful beyond simple correlations. Authors should provide these evaluations.

 As suggested, we calculated the effect sizes for clinical variables between two groups and of analysis of covariance (ANCOVA) testing.

The study just included 41 participants. The sample set is small for the number of covariates used in ANCOVA. This may lead to overfitting and reduce the statistical robustness of the conclusions. How does overfitting reduce here?

 We agreed with the reviewer’s suggestion, we calculated the effect sizes for analysis of covariance (ANCOVA) testing. We believe that the result reduce overfitting.

The study design does not allow causal inference. Therefore, statements linking RWA to disease severity should be softened or rephrased as associations, not predictors.

 As suggested, we could not find the statements linking RWA to disease severity in the present study and did not use “predictors”. If there is such a phrase in the text, could you please point it out. Moreover, the results of the present study did not show the RWA to disease severity.

The introduction repeats known information about RBD and Parkinson’s disease.
Authors could shorten it and focus on the problem addressed by the study would improve clarity.

According to the reviewer’s suggestion, we shortened and modified the Introduction to address focus on the problem in the present study.

In Methods, need more clarification about the following:

Provide exact criteria and thresholds for RWA scoring.

 As suggested, as described in the Methods of the present version, the EMG activity would be more than 50% occupied on the mini-epoch for 3 seconds when phasic muscle activity lasts between 0.1 and 5 seconds and its amplitude is four times greater than the background. When such EMG activity will be evident, the epoch is defined by RWA.

Clarify how many nights of recording were used.

According to the reviewer’s suggestion, we used the recording results of the portable device on two consecutive nights before the zonisamide intervention on the randomized clinical trial of ZEAL. We added this information in the text.

Describe data quality checks and signal rejection procedures.

As suggested, the data from the ZEAL study that formed the basis of the present study was checked for data quality by the data manager and monitoring members of Institute for Clinical and Translational Science, and principal investigator. Prior to that, we held discussions with the manufacturer, Proassist Co., Japan to check the quality of the Sleep graph data. The data entry procedure was carried out by both the principal investigator and a co-researcher, and the consistency of the entered data was confirmed. These information was cited in the Methods of the text.

The figure or architecture would benefit from clearer labelling and consistent formatting. It should be provided in the methods and results sections.

 According to the reviewer’s suggestion, we did this suggested clearer labelling and consistent formatting in the Methods and Results.

Abbreviations are not given properly.

 According to the reviewer’s suggestion, we carefully checked abbreviations and modified them.

Reviewer 3 Report

Comments and Suggestions for Authors

The study by Hiroshi Kataoka et al. is a post-hoc analysis aimed at distinguishing clinical and demographic characteristics in patients with RWA in PD, based on a clinical trial. The question is interesting, but several methodological aspects need clarification before proceeding further:

- The introduction is too long, whereas it would be important to summarise it and make the new elements and objectives of the study more evident;

- It is unclear where the EMG data that the Authors reasonably used to identify the presence of RWA comes from. Data should be presented in this regard;

- It is unclear what the authors mean by “variations”; was the goal only to distinguish RWA from non-RWA as baseline data? Or were the differences observed after drug treatment also taken into account analyses?

- The disease history was similar between patients with and without RWA, but it is unclear whether this also includes the RBD phase (reasonable prodrome) in the RWA group. This aspect should be clarified and its implications considered;

- The Authors used ANCOVA even though some of the data collected did not have a normal distribution. The implications should be considered;

- Was the group with RWA taking drugs that could affect sleep architecture?

- Did any of the patients also have other disorders, such as PLM or RLS? If so, the implications should be considered;

- In the Discussion, I suggest exploring the advantages of this method over other validated and promising methods, such as actigraphy, by discussing and citing: doi:10.1038/s41746-025-01999-z;

- To suggest the importance of PSG assessments in the differential diagnosis of different types of cognitive decline (not only RBD/PD), discussing and citing: doi:10.3390/jcm14207437.

Author Response

Responded to the reviewer’s suggestions

Reviewer 3

Comments and Suggestions for Authors

- The introduction is too long, whereas it would be important to summarise it and make the new elements and objectives of the study more evident;

According to the reviewer’s suggestion, we modified and summarized the Introduction to make the new elements and purpose of the present study.

- It is unclear where the EMG data that the Authors reasonably used to identify the presence of RWA comes from. Data should be presented in this regard;

As suggested, as described in the Methods of the present version, the EMG activity would be more than 50% occupied on the mini-epoch for 3 seconds when phasic muscle activity lasts between 0.1 and 5 seconds and its amplitude is four times greater than the background. When such EMG activity will be evident, the epoch is defined by RWA. The epoch is defined by RWA. An automatic calculation was made to determine RWA as a proportion of total REM sleep epochs. The proportion is cited as a supplemental material. But according to the reviewer’s suggestion, we changed from supplemental table 1 to Table 4 in the text.

- It is unclear what the authors mean by “variations”; was the goal only to distinguish RWA from non-RWA as baseline data? Or were the differences observed after drug treatment also taken into account analyses?

As suggested, our purpose is to investigate whether RWA on the portable EEG/EOG recording device with two channels at home could be associated with clinical PD variations or other sleep metrics. The goal was not only to distinguish RWA from non-RWA as baseline data. Our used data on ZEAL is baseline date not that after the intervention. The text includes three places (Introduction: last sentence, Study design and participants: line 1, 3th paragraph of the Discussion line 13) of the “variation” and we modified the term to avoid this confusion.

- The disease history was similar between patients with and without RWA, but it is unclear whether this also includes the RBD phase (reasonable prodrome) in the RWA group. This aspect should be clarified and its implications considered;

 As suggested, patients with RWA had not typical clinical history of RBD and dream-enacting behavior in the prodromal stage of PD. We added this information in the Results.

- The Authors used ANCOVA even though some of the data collected did not have a normal distribution. The implications should be considered;

 As suggested, we repeatedly and carefully checked the normal distribution for variables, and noticed that the MDS-UPDRS part 3 does not normally distributed. Also. we noticed that the presence of depression is defined as BDS score≥ 14. Again, we re-performed the statistical analysis including ANCOVA testing. We modified the text and tables.    

- Was the group with RWA taking drugs that could affect sleep architecture?

As suggested, three patients in the group with RWA received etizolam. We added this information in the Results.

- Did any of the patients also have other disorders, such as PLM or RLS? If so, the implications should be considered;

As suggested, although all patients did not recieve PSG, all had not periodic limb movements or restless leg syndrome on the clinical routine examination before being registered in the present study. The information was added in the methods and we enhanced the limitation.

- In the Discussion, I suggest exploring the advantages of this method over other validated and promising methods, such as actigraphy, by discussing and citing: doi:10.1038/s41746-025-01999-z;

 Thanks for your suggestion, we added this discussion by use of your suggested reference.

- To suggest the importance of PSG assessments in the differential diagnosis of different types of cognitive decline (not only RBD/PD), discussing and citing: doi:10.3390/jcm14207437.

Thanks for your suggestion again, we added this discussion by use of your suggested reference.

Round 2

Reviewer 2 Report

Comments and Suggestions for Authors

Still, the following suggestions are not addressed.

The figure or architecture would benefit from clearer labelling and consistent formatting. It should be provided in the methodology and results sections.

The first letter in the full form of abbreviations should be CAPS.

The conclusion should not contain a citation because it is a summary of the research work.

Author Response

The figure or architecture would benefit from clearer labelling and consistent formatting. It should be provided in the methodology and results sections.

  According to the reviewer’s suggestion, we more added labelling of Tables in the Methods and Results. If there is a labelling in the methodology and results sections, Please let me know where need the part of the Methods and Results 

The first letter in the full form of abbreviations should be CAPS.

  According to the reviewer’s suggestion, we modified abbreviations.

The conclusion should not contain a citation because it is a summary of the research work.

  According to the reviewer’s suggestion, we delated the citation from the Conclusion.

Reviewer 3 Report

Comments and Suggestions for Authors

I thank the Authors for their work, which significantly improved the quality of their manuscript. No further comments

Author Response

I thank the Authors for their work, which significantly improved the quality of their manuscript. No further comments

  Thanks very much for your review. Your review was very helpful to modify our manuscript.